# Trait-Based Method of Quantitative Assessment of Ecological Functional Groups in the Human Intestinal Microbiome

**DOI:** 10.3390/biology12010115

**Published:** 2023-01-11

**Authors:** Andrew I. Kropochev, Sergey A. Lashin, Yury G. Matushkin, Alexandra I. Klimenko

**Affiliations:** 1Institute of Cytology and Genetics, Novosibirsk 630090, Russia; 2Kurchatov Genomic Center of ICG SB RAS, Novosibirsk 630090, Russia; 3Department of Natural Sciences, Novosibirsk State University, Novosibirsk 630090, Russia

**Keywords:** trait-based approach, functional groups, microbial ecology, human gut microbiome

## Abstract

**Simple Summary:**

Understanding what functions can be performed by members of the human gut microbial community and how they are interconnected can be very useful for the comprehension of this ecosystem’s function in human health and disease. Here, we propose an original approach to derive this information from metatranscriptomes and test it on publicly available data. The main idea is to develop an ecosystem-centric method aimed at quantitatively assessing the activity of particular groups of microorganisms associated with crucial functions performed by the human gut microbiota, such as the production of butyrate and acetate, the reduction of sulfate, and the decomposition of mucin—the key component of the intestinal mucus layer. The proposed method provides more information about the structure and properties of the analyzed ecosystem than other similar methods. We believe that such a strategy has great potential for biomedical research and potential applications in clinical medicine.

**Abstract:**

We propose the trait-based method for quantifying the activity of functional groups in the human gut microbiome based on metatranscriptomic data. It allows one to assess structural changes in the microbial community comprised of the following functional groups: butyrate-producers, acetogens, sulfate-reducers, and mucin-decomposing bacteria. It is another way to perform a functional analysis of metatranscriptomic data by focusing on the ecological level of the community under study. To develop the method, we used published data obtained in a carefully controlled environment and from a synthetic microbial community, where the problem of ambiguity between functionality and taxonomy is absent. The developed method was validated using RNA-seq data and sequencing data of the 16S rRNA amplicon on a simplified community. Consequently, the successful verification provides prospects for the application of this method for analyzing natural communities of the human intestinal microbiota.

## 1. Introduction

The human gut is a habitat for a diverse and complex microbial ecosystem of trillions of bacteria. Members of the microbial community can interact with their host in a variety of ways—from establishing symbiotic relationships to participation in the pathogenesis of numerous diseases such as diabetes, obesity, and other gut-related disorders [1,2,3]. The way these interactions are structured largely depends on the relationships between these bacteria and the structure of a microbial ecosystem.

The multi-omics approaches characterize the entire community in different ways, allowing us to assess the diversity of species [4,5], their functional potential [6], and gene expression [7]. However, the investigation of microbial relationships in natural communities using these approaches faces certain obstacles. It is rather difficult to distinguish patterns in such data and to draw mechanistic conclusions about the processes of interest since, in large systems, the effect of these processes can overlap with a large number of other unknown, uncontrolled, and simultaneously acting factors.

All these facts push researchers to focus on synthetic microbial communities to study ecological processes [8]. Such communities are usually designed using a bottom-up approach, where a limited number of microbial populations are used to assemble a simplified community inhabiting a well-characterized and controlled environment. This allows one to gradually increase the complexity of the studied ecosystems and track the impact of each additional element added to it, ensuring high controllability and reproducibility. As a result, such a community serves as a proper model for natural ecosystems, which helps to understand the fundamental principles of metabolite-mediated ecological interactions [9,10]. Moreover, synthetic microbial communities are exceptionally useful for developing and calibrating new methods of investigation of microbial ecosystems.

Another problem that hinders studying the role of bacteria in ecosystems is the difficulty of assigning genes and their products to particular species. First, not all representatives of the human microbiota receive high-quality reference genomes [11]. Second, the widespread occurrence of horizontal gene transfer further complicates this problem because a gene can be introduced from another phylogenetic group [12]. Despite the fact that this problem can be partially solved using the approaches based on k-mer analysis [13], it remains far from being resolved.

To avoid these problems and achieve a thorough mechanistic understanding of ecosystem processes, it seems relevant to use trait-based approaches [14]. They focus on the quantitative assessment of phenotypic characteristics that affect the population performance through environmental gradients, regardless of species. Such approaches do not require precise taxonomic identification of organisms and rather focus on their ecological functional roles. They represent the characteristics of organisms in terms of their numerous biological attributes, such as physiological, morphological ones, or any other quantitative traits. Recently, trait-based approaches have become widespread and even adopted for describing microbial ecosystems [15,16,17]. In modern molecular genetics and bioinformatics methods, we rarely deal with phenotypic data but rather with indirect information about it. However, a huge amount of information that is being accumulated using new methods such as metagenomics, metatranscriptomics, and metaproteomics can also be employed for the development of theoretical rethinking and adaptation of the trait-based approaches derived from macroecology.

We use the trait-based approach to develop a method for dissolving the problem of lacking taxonomic identity information by focusing on ecologically significant quantitative traits, presumably involved in a phenotype formation. We identify the entire set of genomic data features that contain information about the phenotypical traits of bacteria relevant for ecological interactions.

Modern functional analysis methods such as HumanN and Karnelian [18,19] aim to analyse the entire metagenome or the metatranscriptome as a whole and represent all known metabolic pathways. However, such tools yield information that is redundant for the reconstruction of environmental relations. We believe that broad strategies inevitably incur certain costs, such as the unnecessary consumption of computing resources and difficulties in formalizing a quantitative assessment of feature abundances.

Our approach, linking the data from high-throughput sequencing to the structure and properties of the ecosystem, aims to characterize environmental relations accurately, which is important for biomedical research and potential application in clinical medicine.

## 2. Methods

### 2.1. Metagenomic and Metatranscriptomic Data from a Synthetic Community Modelling the Core Microbiome of the Human Gut

We chose the gut microbial community described in the article [10] as the object of our study. It consists of 14 species of sequenced commensal bacteria that perform the core metabolic activities of human gut microbiota. The authors carried out gnotobiotic mice colonization by this community and used it as a model to study the links between dietary fiber content, microbiota composition, and mucus layer health.

In our method, we use transcriptome data from experiment 1 of the article obtained from the cecum at day 54. Throughout the experiment, mice were fed a variety of diets, including a fiber-rich diet (R)—four mice, a fiber-free diet (F)—three mice, and the alternation of a fiber-rich diet and fiber-free diet (FR)—six mice. At the end of the experiment, the mice were sacrificed, and the intestinal contents were sent for further analysis. For all mice, the abundance of bacteria in the gut microbiota was measured based on 16S rRNA, transcriptome analysis was performed only for three mice per diet selected at random.

We estimated the adequacy of our reconstruction by comparing the functional group’s abundance with the bacterial taxonomic abundance based on 16S rRNA obtained in that paper (see Appendix A) [10]. Figure 1 shows how bacterial taxa were grouped into functional groups for comparison. Thus, the independent information is used to verify the method, which is not involved in its construction. All the used data are described in Appendix A.

### 2.2. Methodology for Identification of the Key Functional Groups and Their Trait-Determining Genetic Features (TDGFs)

To implement a trait-based method for quantifying the activity of functional groups in the human gut microbiome based on metatranscriptomic data, it is necessary to distinguish the functional groups in the microbial community (see Figure 1 and more details in Appendix A). Under a functional group, we mean a set of organisms that obtain the metabolic activity of interest that affects the entire community. We reconstructed metabolic interactions between the community members using an expert search and analysis of the literature [10,20,21,22,23,24,25,26,27,28,29,30,31,32,33,34,35,36,37,38,39].

To assess the abundance of each functional group, we introduce the concept of trait-determining genetic features (TDGF). It is a particular genetic sequence for a key enzyme, which is usually responsible for a bottleneck in the metabolic pathways of interest (more details in Appendix A). Under the function that defines a functional group, we mean both a simple metabolic process and a set of various metabolic reactions associated only with a common substrate or product. Quantitative assessment of functional trait occurrence can be expressed through the abundance of a single TDGF (one gene).

A total of eight TDGFs were selected (more details in Appendix A). Carbon monoxide dehydrogenase, acetyl-CoA synthetase, and both corrinoid iron-sulfur protein subunits are involved in the formation of acetyl-CoA in acetogenesis. The dissimilation type sulfite reductase is responsible for the conversion of sulfite to hydrogen sulfide, which is the final stage of the sulfate reduction pathway; butyryl-CoA dehydrogenase catalyzes the key reaction in the formation of butyrate. Sulfatase and Alpha-N-acetylgalactosaminidase break important covalent bonds of mucin.

In cases where several TDGFs could be selected for one functional group, we used the most specific one. It was verified by aligning the TDGF’s sequence to the reference genomes of the representatives of functional groups (more details in Appendix A).

### 2.3. Implementation of the Method for Quantitative Assessment of Functional Groups in the Human Intestinal Microbiome

The presence of the selected TDGFs in the genomes of the representatives of functional groups was verified by aligning their amino acid sequences against the reference genomes of representatives of the microbial synthetic community.

De novo assembly of the transcriptomes was performed using the Trinity platform [40]. To estimate the abundance of transcripts, we used the Kallisto program [41]. Pseudo-alignments were carried out for each sample per previously indexed assembly. The default parameters were used throughout. The data on the abundance of transcripts are presented as TPM values [42].

All single TDGFs were aligned against transcriptome assembly. Since each sequence can align against several contigs, all contigs with score values for alignment less than 250 were excluded from further analysis. This threshold filters out the contigs with low identity. TPM values for contigs containing TDGFs were selected from Kallisto’s output obtained on the previous work steps.

Total TPM values for a TDGF were calculated by summing all TPM values of the TDGF-containing contigs separately for each mouse for different diets. The following single TDGFs were used for identifying respective functional groups: acet1, sulfat1, but1, and muc2 (see Appendix A).

To adapt the estimates of taxonomic abundance for comparison, we used 16S rRNA-based relative abundance values for community members from [10]. Functional groups’ abundances were calculated in similar ways for each mouse, the summation was carried out by the relative abundances of the representatives of these groups. Then, both values were combined into the samples according to the diets, and the average was calculated (see Formulas (1) and (2) in Table 1).

In addition, we calculated the relative proportion of abundance of each functional group in relation to the total abundance of all the distinguished functional groups (see Formulas (3) and (4) in Table 1).

All scripts that link various stages of the work, as well as the scripts required for the last stages of the analysis, namely filtration and summation, were written in the Python programming language (see Appendix A.

### 2.4. Comparing Functional Analysis Results with Humann

The analyzed data were processed within HumanN 3, following its standard protocol [18]. We have regrouped the output file, which contains the list of UniRef gene families and their abundance estimates, to obtain sequence abundance estimates for the corresponding EC (Enzyme Commission) numbers. Further, the CPM values were processed in the same way as it is described in the previous section.

## 3. Results

### 3.1. The Ecological Structure of the Human Intestinal Microbial Community in Terms of Functional Groups

We integrated information about the structure of the ecosystem from the literature into the network of metabolic interactions. It includes information on the metabolism of glycans, long-chain fatty acids, amino acids, sulfates, and various other metabolites (see Appendix A). To select functional groups describing key metabolic processes important for the functioning of the gut microbiota-host system, we settled our choice on four functional groups: butyrate-producers, acetogens, sulfate-reducers, and mucin-decomposing bacteria. The division of the community into functional groups and the most important ecological interactions are summarized in Figure 1.

We formulated hypotheses about how the abundance of functional groups will change depending on diets and assessed the consistency of the developed method while obtaining additional information about the functioning of the community, which had not been previously discussed in [10].

It is known that under the conditions of switching to a fiber-rich diet, the bacteria decompose fibers, which reduces the intensity of degradation of the mucin layer [10]. This could cause a decrease in the concentration of sulfate in the system, and sulfate-reducing bacteria are suppressed. In turn, the suppression of sulfate-reducers could lead to the dominance of acetogens and to an increase in acetate production.

Since acetogens require a higher concentration of hydrogen for their activity compared to sulfate-reducers [35], a decrease in the abundance of the latter could lead to an increase in the concentration of hydrogen in the system, inhibition of fermentation, and an overall decrease in the production of short-chain fatty acids including butyrate [39]. Additional production of acetate by acetogens could stimulate the production of butyrate due to its conversion from acetate. However, this effect is unlikely to be significant since acetate is mainly formed as a product of the decomposition of carbohydrates, and not due to the relatively small number of acetogens in natural communities [20].

Thus, our reconstruction integrates the accumulated knowledge of the metabolic interactions of bacteria in the studied community and assists further implementation and verification of our method.

We chose TDGFs that allowed us to assess the abundance of functional groups for the high-throughput sequencing data of intestinal microbiota communities. Since we exactly know from the literature the affiliation of the representatives of the synthetic microbial community to functional groups, we verified that the TDGFs were chosen correctly by aligning each TDGF against the available reference genomes of the studied microorganisms (see Appendix A).

### 3.2. The Similarity of the Patterns of Changes in the Abundance of Functional Groups When Evaluated Based on Different Data

The implementation of the developed method provided us with data on changes in the abundance of TDGFs depending on three types of diets: fiber-free diet, fiber-rich diet, and a diet where dietary regimes alternated.

The abundance of functional groups has been assessed based on RNA-seq data and 16s rRNA amplicon data. Figure 2A–D shows the comparison of the relative abundance of functional groups based on 16S rRNA sequencing data with the abundance of TDGFs of these functional groups based on RNA-seq data (Formulas (1) and (2) were used). Comparing these two methods for degrading mucin under different diets, we clearly see that only a fiber-rich diet leads to a decrease in the activity of mucin-decomposing bacteria (Figure 2A). This point fully confirms the results that were obtained by [10], which indicates the correctness of our choice of the TDGF for this functional group.

We see a roughly similar situation for sulfate reducers (Figure 2B). The presence of their TDGF is greatly reduced only in a fiber-rich diet. This is primarily because the decomposition of mucin releases sulfate, which leads to an increase in the activity of this functional group.

If we analyse the abundance of acetogens depending on diets, we see that there is a significant similarity of the pattern (Figure 2D). However, the situation is not so clear-cut for switching from an alternating diet to fiber-rich diet. This may be attributed to acetogenesis being mostly an optional function for acetogens, and the conditions of the fiber-rich diet presumably do not facilitate the transcription of enzymes for this function. However, the confidence intervals for these values are large and greatly exceed the difference between the mean values, and the observed pattern of changes in values is not significant (the *p*-value for the Welch test is 0.52). In general, the pattern of changes in values is consistent with our initial assumptions that acetogenesis will be more active in the absence of sulfate reduction.

The situation is quite interesting for butyrate-producing bacteria (Figure 2C). The data on the relative abundance of microorganisms based on 16S rRNA sequencing show that the relative proportion of butyrate-producing bacteria increases dramatically when switching to a fiber-rich diet. However, the TPM values for this functional group decrease under the same conditions. The probable reason for the observed discrepancy may be the difference between the species abundance and the actual expression of the corresponding TDGF. The taxonomic abundance of butyrate-producers does not guarantee the expression of the corresponding TDGF. Thus, a change in the expression may occur, with a shift in fermentation products in favour of other short-chain fatty acids.

Interestingly, total TPM values decrease when shifting to a fiber-rich diet for all TDGFs taken together (Figure 2E). Apparently, fiber-rich nutrition favors bacteria to use their metabolic potential more actively; therefore, the share of our TDGFs decreases compared to the overall expression.

The estimates of functional groups’ abundance, expressed in terms of TPM values and in terms of relative abundance, mainly correlate with each other. However, in order to compare these methods adequately, additional normalization is needed. We compared the proportions of each functional group with all available functional groups, ignoring the unclassified. This method neutralizes the effect that the expression of different transcripts can vary according to environmental conditions. This effect can lead to errors in situations where, for example, depending on changes in environmental conditions, the proportion of transcripts expressed by the majority of bacteria in the community increases. To avoid this, we have estimated the abundances according to Formulas (3) and (4) (see Section 2).

These two methods for each functional group determine its relative proportion among the rest of the functional groups in a different way (see Figure 3) because the taxonomic abundance of functional groups does not guarantee the corresponding expression of their TDGFs. The patterns of changes in the abundance of acetogens, sulfate reducers, and mucin-decomposing bacteria correlate well between the two different quantification methods. The exception is butyrate-producing bacteria—the correlation is weaker due to the large range of the confidence intervals.

### 3.3. Comparing Functional Analysis Results with HumanN and RSEM

We also compared our trait-based quantification method (TBQM) with another widespread functional analysis method—HumanN 3 [18]. In general, the profiles of changes in values depending on the diet coincide (see Figure 4), however, the HumanN values are approximately ten times lower than those obtained by our trait-based method for each of the TDGFs, which is unexpected and requires an explanation.

Despite the apparent similarities, these two methods are very different in their internal structure. HumanN performs read alignment and mapping using bowtie2 against pre-engineered, functionally-annotated pangenomes of strains that were previously identified using MetaPhlAn2 [43]. Then, for reads that did not match the pangenomes, a translated search was performed in the complex protein UniRef database [44]. In our pipeline, alignments are performed against the assembly of the transcriptome, and Kallisto does not carry out mapping, which ensures the high speed of this algorithm.

To control for a possible bias introduced by different quantification methods, we decided to replace Kallisto with RSEM [45]. RSEM uses bowtie2 as HumanN does, so the implementation of the trait-based quantification method in this mode is closer to HumanN. However, the differences between TPM values when using Kallisto and RSEM are minimal (see Figure 4).

We assumed that HumanN probably ignores some of the reads that might align against the TDGF sequences. To test this hypothesis, we analyzed the SAM files obtained as a result of the pipelines in order to obtain information about the mapping location of the probably ignored reads in the HumanN databases. We found a large number of Uniref90 sequences annotated with the function of TDGFs, and only a relatively small number of uncharacterized sequences were found (for more details see Appendix A). To conclude, the reason for the differences in abundance estimates between the two compared methods is that HumanN provides an incomplete list of Uniref90 identifiers for the EC numbers that are of particular interest for quantifying the activity of the functional groups investigated in this study.

The two methods also differ in their performance bottlenecks. For HumanN it is a translated search against the complex protein UniRef database [44] that is the most computationally demanding procedure. For the trait-based quantification method, the most arduous part of the work is assembling the transcriptome, which is about two times faster.

However, these two methods can be compared only to a certain extent since, initially, they have quite different aims and scopes. HumanN is designed as an exploratory functional analysis tool in the field of metagenomics and metatranscriptomics, and, therefore, it has a much broader application and uses most of the known enzymatic functions. Contrariwise, the method presented in this study has a completely different conceptual foundation. It suggests focusing on functional groups and filtering the information that is relevant to the interactions between the major agents of the microbial community. Although, at the moment, the set of functional groups is quite narrow, the method has a wide space for its further development, taking into account the complexities of the human gut microbial ecosystem.

## 4. Discussion

Trait-based approaches have been actively spreading in microbial ecology recently [15,46]. However, a number of problems complicate their use by researchers. Such approaches have been adopted from macroecology, where the traits of individual organisms can be easily observed and measured, while microbial ecologists are significantly limited in such possibilities [47].

The advent of high-throughput sequencing has provided huge data on microorganisms, but the phenotype can be predicted only indirectly on their basis [48]. Meanwhile, there is no standardized definition of a microbial functional trait and a protocol for their measurement, and researchers choose the traits depending on their specific interests and their experience [47].

In our study, we aimed to address this problem by assigning TDGFs to traits and measuring their abundance at the transcript level. The latter is especially important since metatranscriptomic data, in comparison with metagenomic data, contain information on the expression of individual genes and allow for a better assessment of community functions. Thus, we provide a basis for quantifying the abundance of traits expressed in the microbial community.

The verification of the developed method showed that the quantitative assessment of changes in the functional structure of the ecosystem is consistent with the conclusions of [10]. For instance, we detect a decrease in the functional abundance of mucin-decomposing bacteria when switching to a fiber-rich diet, which is in accordance with the conclusions about the positive role of complex plant fibers made in the article.

Moreover, our hypotheses about how changes in the abundance of functional groups should occur depending on the change in diets are indirectly confirmed. Thus, the closest connection was found to be between the functional group of mucin-decomposing bacteria and sulfate-reducers: any decrease in abundance of the former coincides with decrease in abundance of the latter. Apparently, the resource limitation on sulfate, which sulfate reducers experience with a decrease in the abundance of mucin-decomposing bacteria, is very strict.

The inverse relationship between the abundance of sulfate-reducers and acetogens is also largely observed, with the exception of switching from alternating to a fiber-rich diet. Since acetogenesis is a facultative function for *Marvinbryantia formatexigens* [35], it can be assumed that acetogenesis in such conditions reduces its role in general metabolism.

Additionally, our method showed a general decrease in the activity of all selected TDGFs under a fiber-rich diet, which is most likely associated with the activation of transcription of other genes and an increase in the diversity of functions performed by the microbiome. This observation is of great interest, and the literature has repeatedly discussed the relationship between taxonomic diversity, diet, and health [1,2,3]. However, it has not yet been shown that an increase in functional diversity can occur due to changing diets in a community where the taxonomic composition is completely the same.

In addition, the decrease in TPM values for the selected TDGFs is associated with a discrepancy in taxonomic and functional changes in abundance, calculated using formulae without normalization by the selected functional groups. The activity of functional groups decreases while switching to a fiber-rich diet, the only exception is acetogens when switching from a fiber-free diet to a mixed diet. This situation necessitates being very careful when comparing such data and keeping in mind that their compositional and heterogeneous nature can significantly affect the analysis results.

There are some prospects and challenges, such as a more accurate assessment of the functional groups’ abundance in natural communities considering all the variety of sequences that are responsible for the marked function. The chosen threshold for alignment on the sequence of TDGFs should be adjusted for natural communities to reduce false results. Moreover, there are also sequences of non-homologous, isofunctional enzymes that can duplicate the functions of TDGFs, while their nucleotide sequences are radically different [49]. Further development of the method involves expanding the sequence catalogue for each TDGF, which can largely solve the mentioned problems.

We used single TDGFs for the functional groups throughout the study, however, if there are multiple TDGFs for the process of interest, the following question arises: How does one derive an activity of a functional group from TPM values of different sequences? Since the contribution of each of the TDGFs to the expressed function might differ, it complicates assessment of functional group abundances in the case of multiple TDGFs. Using single TDGFs for functional groups allows us to bypass this problem but limits us in distinguishing more complex functional groups; for example, we cannot assess the total activity of all different specialists in the decomposition of complex polysaccharides, which also play an important role in the community. Besides, we used only one TDGF for mucin-decomposing bacteria, while there is a wide range of enzymes that are involved in its degradation. Undoubtedly, considering the contribution of all this diversity of enzymes would increase the accuracy of assessing the activity of the functional groups.

## 5. Conclusions

To conclude, the developed method has bright prospects for the analysis of metatranscriptomic data. Its verification was carried out on the synthetic community, and the results of the analysis of changes in the ecological community structure under various dietary regimes are in accordance with the previously published results and with the knowledge concerning functioning of the human intestinal community discussed in the scientific literature. Thus, the developed method enables using metatranscriptomic data for a functional analysis of the human gut microbiome and thereby assessing quantitatively the activity of the functional groups in this ecosystem.

## Figures and Tables

**Figure 1 biology-12-00115-f001:**
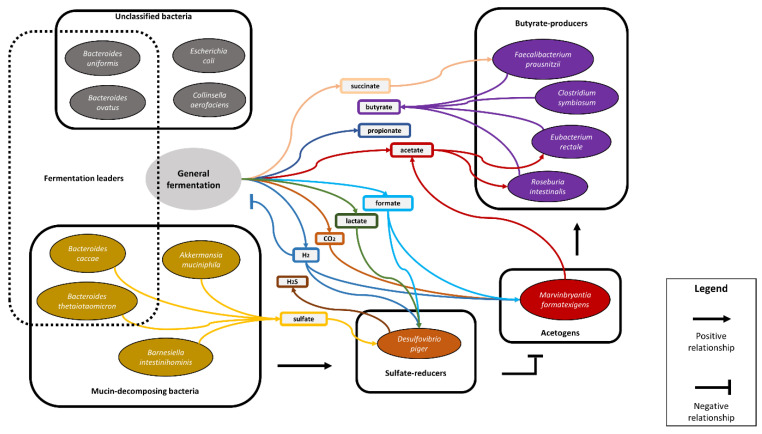
General scheme of the reconstructed metabolic interaction network. Outcoming arrows denote the production of metabolites, incoming arrows denote their consumption by the members of respective functional groups. The production of metabolites that results from the metabolic activity of most organisms is referred to as “general fermentation”.

**Figure 2 biology-12-00115-f002:**
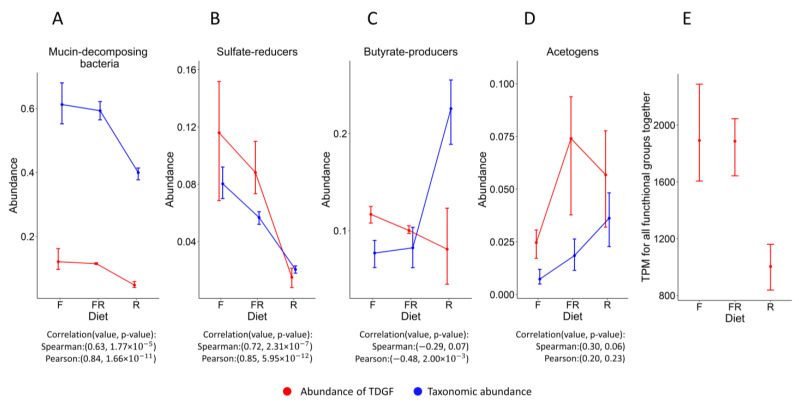
Comparison of the abundance of taxonomic groups. F—fiber-free diet, R—fiber-rich diet, FR—alteration between fiber-free and fiber-rich diets. (**A**–**D**)—Comparison of TPM values calculated based on RNA-seq data and relative abundance of functional groups based on 16S rRNA sequencing data. TPM values are reduced by 5000 times for the ease of visualization in one figure. The correlation between the abundance estimated by two different methods is calculated. (**E**)—total TPM value for all TDGFs together for different types of diets.

**Figure 3 biology-12-00115-f003:**
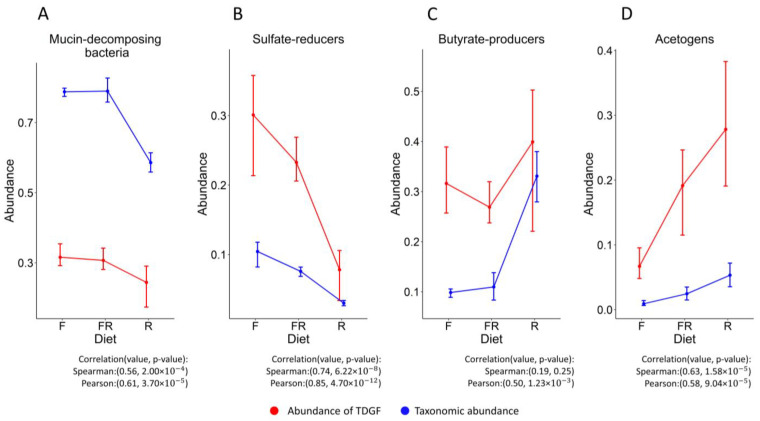
Comparison of TPM values calculated based on RNA-seq data and relative abundance of functional groups based on 16S rRNA data adapted according to the Formulas (3) and (4). F—fiber-free diet, R—fiber-rich diet, FR—alteration between fiber-free and fiber-rich diets. Acet1 was used as the TDGF of acetogenesis. The correlation is calculated for the corresponding values between the two methods.

**Figure 4 biology-12-00115-f004:**
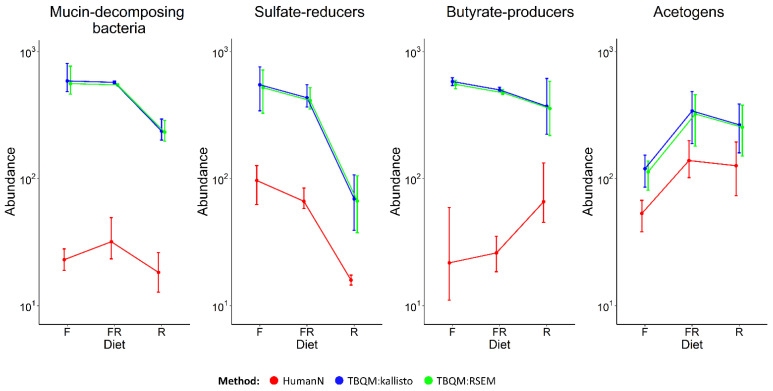
Comparison of TPM values of functional groups for the TBQM (with Kallisto and with RSEM) and CPM values for HumanN. F—fiber-free diet, R—fiber-rich diet, FR—alteration between fiber-free and fiber-rich diets.

**Table 1 biology-12-00115-t001:** Formulae for assessing the abundance of functional groups classified according to used quantification and normalization methods. *Fg* is the analyzed functional group, *M* is the TDGF selected for analysis, *B* is the bacteria that belong to the analyzed functional group, *C* is the isoform of the transcript to which TDGF aligned, *D* is the diet, *E* is the mouse (separate experiment).

		**Quantification Method**	
		*Abundance of the functional group based on* TDGF *identification*	*Abundance of the functional group based on taxonomic identification (16S rRNA)*
**Normalization method**	*Its proportion to the whole community*	$\frac{1}{k}\overset{k}{\underset{j=1}{\sum }}\overset{m}{\underset{i=1}{\sum }}TPM\left(M,D,C_{i},E_{j}\right)\;\left(1\right)$	$\frac{1}{k}\overset{k}{\underset{j=1}{\sum }}\overset{m}{\underset{i=1}{\sum }}16SrRNA\left(Fg,D,B_{i},E_{j}\right)\;\left(2\right)\;$
	*Its proportion to the total abundance of the distinguished functional groups*	$\frac{1}{k}\overset{k}{\underset{j=1}{\sum }}\frac{\sum_{i=1}^{m}TPM\left(M,D,C_{i},E_{j}\right)}{\sum_{j=1}^{k}\sum_{i=1}^{m}TPM\left(M_{j},D,C_{i},E_{j}\right)}\;\left(3\right)$	$\frac{1}{k}\overset{k}{\underset{j=1}{\sum }}\frac{\sum_{i=1}^{m}16SrRNA\left(Fg,D,B_{i},E_{j}\right)}{\sum_{j=1}^{k}\sum_{i=1}^{m}16SrRNA\left(Fg_{i},D,B_{i},E_{j}\right)}\;\left(4\right)$

## Data Availability

The analyzed data were taken from [10] and have the following BioProjectID identifiers: SRP092478, SRP092476, SRP092461, SRP092458, and SRP092453. All other data generated or analysed during this study are included in this published article and its supplementary information files. The code is available at https://github.com/Yasana1990/metacrest accessed on 1 January 2022 and also located in Appendix A.

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
