# Peer review of "Trait-Based Method of Quantitative Assessment of Ecological Functional Groups in the Human Intestinal Microbiome"

_biology, 2023, doi:10.3390/biology12010115_

Round 1

Reviewer 1 Report

The authors of the work entitled “Trait-based method of quantitative assessment of ecological 2 functional groups in the human intestinal microbiome” performs a meta-analysis of already published data to linking the data from high-throughput sequencing to the structure of the microbiome, to characterize ecological relations.

Just some minor comments:

The link to the code is not working, the correct should be: https://github.com/Yasana1990/metacrest

Author Response

Dear Reviewer,

We are glad that you appreciated our manuscript. As for the link to the code, we have fixed it in the current revision.

Sincerely yours,

Andrew Kropochev on behalf of the author's collective.

Reviewer 2 Report

The manuscript by Andrew I. Kropochev and colleagues focuses on transcriptomic trait important in functionally defining microbial communities, at least for four defined trait. Interesting and sounding the comparison with the state-of-the-art HUMAnN3 pipeline, which is usecfor general purposes in metagenomics and metatranscriptomics, but it is well known for it non-completeness. No special amenments are neede dto the mansuxript in teh present form, but I would like to suggest Authors to try broadening their 4 set of selected single TDGFs, for example adding bacterial AMR genes and/or detoxifying genes: it would be interesting to see if a fiber-rich diet would diminish or enhance these other two gene(s) traits. Very informative also the set of Supplementary Information files, appreciated for their completeness and reliability for the reader. Little ameliorations should be made for the English language.

Author Response

Dear Reviewer,

We thank you for your appreciation of our manuscript. Your proposal to add two new TDGFs is very interesting. In the future, we plan to significantly expand the existing catalog and therefore any advice regarding new TDGFs is of great value to us. As you stated, no special amendments are needed to the manuscript in the present form and we agree because we feel that to this point the manuscript represents an accomplished study, which paves the way for the future research in these directions.

Sincerely yours,

Andrew Kropochev on behalf of the author's collective.

Reviewer 3 Report

Dear author, 

Herein you propose an interesting procedure without validating It in the Real Life.

I do not appreciate this type of manuscript. Thus i'm obliged to suggest the rejection.

Author Response

Dear Reviewer,

Such a categorical assessment of our manuscript seems to be a bit strange for us. The work described by us is indeed bioinformatics and the validation of the presented method relies on previously published data. However, it seems to us that this is a very good practice when the data obtained earlier can be used in a new way. If your remark is more related to the nature of the data analyzed in the work, then we should note that the use of synthetic communities provides many advantages for developing and testing new biological/bioinformatic methods. This is precisely what the present study design explains. However, we are enthusiastic about extending the developed method and applying it in the future to data from other communities that may already be observed in the Real Life.

Sincerely yours,

Andrew Kropochev on behalf of the author's collective.
